# Tangent-space methods for truncating uniform MPS

Bram Vanhecke,[1, *] Maarten Van Damme,[1] Jutho Haegeman,[1] Laurens Vanderstraeten,[1] and Frank Verstraete[1]

[1]*Department of Physics and Astronomy, University of Ghent, Krijgslaan 281, 9000 Gent, Belgium*

A central primitive in quantum tensor network simulations is the problem of approximating a matrix product state with one of a lower bond dimension. This problem forms the central bottleneck in algorithms for time evolution and for contracting projected entangled pair states. We formulate a tangent-space based variational algorithm to achieve this for uniform (infinite) matrix product states. The algorithm exhibits a favourable scaling of the computational cost, and we demonstrate its usefulness by several examples involving the multiplication of a matrix product state with a matrix product operator.

The density matrix renormalization group (DMRG)[1,2] and quantum tensor networks[3,4] provide algorithms for simulating ground states of strongly correlated quantum many body systems with a computational cost that scales linear in the system size, thereby overcoming the infamous exponential wall of the quantum many body problem. The physical parameter controlling the computational cost is the entanglement entropy, as directly reflected in the bond dimension $\chi$ of the corresponding matrix product states (MPS)[5]. However, there are many interesting physical problems for which this bond dimension can become prohibitively large, such as the problem of simulating time evolution of a quantum state out of equilibrium or of contracting a tensor network comprised of a projected entangled pair state (PEPS) with a large bond dimension. In both cases, the central problem is to approximate the product of an MPS and a matrix product operator (MPO) with a MPS of lower bond dimension. For both finite and infinite systems, a well known algorithm to achieve this is time-evolving-block-decimation and variants thereof[6–9]. For finite systems, a considerable improvement over those algorithms can be obtained by adopting a variational algorithm which optimizes the fidelity by sweeping through the system while solving alternating linear problems[10,11]. The computational cost of the latter algorithm has a better scaling as it does not require to bring the joint MPS/MPO system in canonical form, and furthermore achieves a better overal fidelity due to its variational nature.

In this paper, we present the uniform and infinite version of that algorithm. It is based on ideas developed in the context of tangent space methods for uniform matrix product states[12,13] and the variational uniform matrix product state algorithm[14,15]. Our main motivation is the development of efficient MPS algorithms which can deal with time-evolution methods involving MPOs with large bond dimension and of efficient and well-conditioned ways of contracting PEPS[16]. It also overcomes a main limitation of algorithms based on the time-dependent variational principle (TDVP)[17–19], where it is difficult to build up entanglement starting from a low-entangled state by allowing large time steps.

The paper is organized as follows. In the first section we discuss how to approximate a given uniform MPS variationally with another with with lower bond dimension.

In a second section, we illustrate this algorithm with several relevant examples.

*Fixed-point equations.*—We start from the diagrammatic expression of a uniform MPS in the thermodynamic limit, parametrized by a single tensor $A$

$$|\Psi(A)\rangle = \ \text{—}\!\!A\!\!\text{—}\!\!A\!\!\text{—}\!\!A\!\!\text{—}\!\!A\!\!\text{—}\!\!A\!\!\text{—} \ . \qquad (1)$$

We will assume a trivial unit cell in this text for simplicity, the case of larger unit cells is treated straightforwardly. Using the gauge freedom of the MPS we can choose this tensor to be in the left canonical gauge $A_L$ or the right canonical gauge $A_R$, with

$$\qquad (2)$$

These gauge-fixed tensors are related by a matrix $C$ as

$$\text{—}\!A_L\!\text{—}\!C\!\text{—} \ = \ \text{—}\!C\!\text{—}\!A_R\!\text{—} \ = \ \text{—}\!A_C\!\text{—} \ , \qquad (3)$$

allowing us to bring the MPS into the so-called mixed gauge

$$|\Psi(A)\rangle = \ \text{—}\!A_L\!\text{—}\!A_L\!\text{—}\!A_C\!\text{—}\!A_R\!\text{—}\!A_R\!\text{—} \ . \qquad (4)$$

For a given MPS $|\Psi(M)\rangle$ described by a tensor $M$, we now wish to find an MPS $|\Psi(A)\rangle$ such that the latter approximates the former in some optimal way. A natural choice for an optimality condition is that they should have a maximal fidelity, which leads us to a variational optimization problem for the tensor $A$

$$A = \arg\max_A \frac{\langle\Psi(A)|\Psi(M)\rangle \langle\Psi(M)|\Psi(A)\rangle}{\langle\Psi(A)|\Psi(A)\rangle}. \qquad (5)$$

This cost function being a real-valued function of $A$ and $A^*$, the gradient is obtained by differentiating the cost function with respect to $A^*$. An optimal point is reached when the gradient vanishes,

$$\langle\partial_{A^*}\Psi(A)| \left( |\Psi(M)\rangle \right.$$
$$\left. - \frac{\langle\Psi(A)|\Psi(M)\rangle}{\langle\Psi(A)|\Psi(A)\rangle} |\Psi(A)\rangle \right) = 0. \qquad (6)$$

The left-hand side of this equation can be interpreted as a tangent vector on the manifold of MPS[12,13], and the optimality condition can be reformulated as

$$\mathcal{P}_A \left| \Psi(M) \right\rangle = \frac{\left\langle \Psi(A) | \Psi(M) \right\rangle}{\left\langle \Psi(A) | \Psi(A) \right\rangle} \mathcal{P}_A \left| \Psi(A) \right\rangle , \qquad (7)$$

where $\mathcal{P}_A$ represents the projector on the space of tangent vectors to $\left| \psi_A \right\rangle$. An explicit form of the tangent-space projector in the mixed-gauge is given by[13]

Applying this operator to $\left| \psi_M \right\rangle$, which we assume to be a uniform MPS parameterized by a single tensor $M$, we find that the optimality condition [Eq. (7)] implies that

$$A_C' = A_L C', \qquad (9)$$

where $A_C'$ and $C'$ are given by

and

with the fixed points $G_L$ and $G_R$ given by the eigenvalue equations

Here, the factor $\lambda$ appears as the 'fidelity per site', formally given in the thermodynamic limit as

$$\lambda = \lim_{N \to \infty} \left( \left\langle \Psi(M) | \Psi(A) \right\rangle \right)^{1/N} . \qquad (13)$$

In order to find a fixed point, we can use an iterative scheme. One crucial step in each iteration will be the

---

**Algorithm 1** Variationally optimizing overlap of uniform MPS with trial state $\left| \Psi(M) \right\rangle$

1: bring $A$ in canonical form $\{A_L, A_R\}$
2: **repeat**
3:     compute $\lambda$, $G_L$ and $G_R$         ▷ Eq. (12)
4:     find new $A_C'$ and $C'$        ▷ Eqs. (10)-(11)
5:     extract new $A_L$ and $A_R$     ▷ Eq. (14)-(15)
6:     compute error $\epsilon$            ▷ Eq. (16)
7: **until** $\epsilon < \eta$
8: **return** $A_L, A_R, \lambda$

---

extraction of a new set of MPS tensors $\{A_L, A_R\}$ from the $A_C'$ and $C'$ that were obtained. A close-to-optimal solution of this problem is given by the prescription[13]

$$A_L \leftarrow U_l V_l^\dagger, \quad \begin{cases} A_C' = U_l P_l \\ C' = V_l Q_l \end{cases} \qquad (14)$$

and

$$A_R \leftarrow U_r^\dagger U_r, \quad \begin{cases} A_C' = P_r U_r \\ C' = Q_r V_r \end{cases} . \qquad (15)$$

where all decompositions involve unique polar decompositions or their transposed. This approach is very similar to the one adopted in the standard variational uniform MPS (VUMPS) algorithm[15]. Once we have obtained a new set $\{A_L, A_R\}$, we can re-compute the fixed-point tensors $G_L$ and $G_R$ and the scheme can be reiterated. As a convergence measure we take the norm of the fixed-point equation in Eq. (7), which is given by

where $A_C'$ and $C'$ are given by Eqs. 10 and 11.

A specific instance of the above scheme occurs when applying a uniform matrix product operator (MPO) to a given MPS, and approximating the resulting state as an MPS with a certain bond dimension. In that case the above fixed-point equations are given by

and

with

$$G_L \left[ \begin{matrix} M \\ O \\ A_L^* \end{matrix} \right] = \lambda \left[ G_L \right] , \qquad \left[ \begin{matrix} M \\ O \\ A_R^* \end{matrix} \right] G_R = \lambda \left[ G_R \right]. \quad (19)$$

Our variational method can, therefore, be used for approximating an MPS-MPO state by an MPS with the original bond dimension of $M$. This is an operation that appears in many MPS methods (see further), and we can show that our approach scales more favourably as compared to the standard local-truncation approach. Indeed, supposing that both the original and new MPS have bond dimension $\chi$ and physical dimension $d$ and the MPO has bond dimension $D$, the time-complexity of the above scheme is $\mathcal{O}(\chi^3 Dd + \chi^2 D^2 d^2)$, and the memory required scales as $\mathcal{O}(\chi^2 Dd)$. We can compare this to the complexity of cutting the bond dimension by truncating local Schmidt values. The most costly operation required to cut the bond this way is following contraction:

$$\rho \left[ \begin{matrix} M \\ O \\ O^* \\ M^* \end{matrix} \right] = \tilde{\rho} \left[ \quad \right] . \quad (20)$$

The time-complexity of this operation is $\mathcal{O}(\chi^3 D^2 d + \chi^2 D^3 d)$ and the memory required $\mathcal{O}(\chi^2 dD^2)$. In addition, one typically performs a full singular-value decomposition of a square $\chi D$ matrix, for which the time complexity scales as $\mathcal{O}(\chi^3 D^3)$. This analysis shows that for MPOs of large virtual dimension $D$, the method we prescribe can be a significant, even crucial, improvement.

*Truncating an MPS.*—Let us first illustrate the method by truncating the bond dimension of a given MPS. The most commonly used technique for that purpose is the truncataction of the Schmidt values on all bonds simultaneously[7]. We compare the two techniques in Fig. 1 for an MPS of considerable dimension. We find that truncating all Schmidt values simultaneously performs fairly well across the board, but that our variational scheme still finds a slightly better state after convergence. This example shows that our fidelity optimization can be useful only if precision is of the utmost importance.

*Time evolution.*—There are roughly two different classes of methods used to time-evolve an infinite MPS. The first class tries to directly transform the Schrödinger equation into a (non-linear) differential equation on the variational manifold. This is exactly the mechanism behind TDVP[17,19], where the direction in which the state needs to change (the right hand side of the Schrödinger equation) is projected onto the tangent space of the MPS.

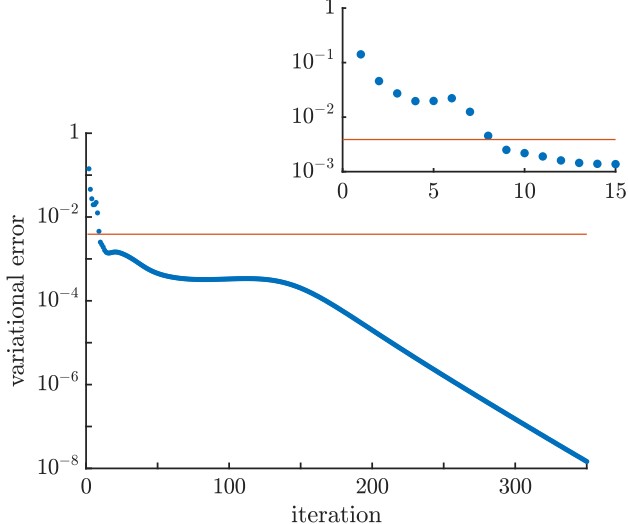

FIG. 1: Truncating an MPS to a lower bond dimension. We show the variational error $\epsilon$ [Eq. (16)] in each iteration of the fidelity optimization (blue), compared to the variational error of the state obtained by local singular-value truncation (red). After eight iterations the variational error is smaller, but we can converge a lot further using our iterative scheme. The fidelity per site $\lambda$ with the original state is $1 - 5.37 \times 10^{-5}$ and $1 - 3.78 \times 10^{-5}$ respectively, showing that we can improve the state with our variational scheme. The starting MPS is an SU(2)-symmetric ground-state approximation for the spin-4 Heisenberg model with 13 charge sectors and maximal bond dimension in each sector $D_{\max} = 512$, yielding a total bond dimension of $D_{\text{total}} \approx 21600$. The truncated MPS has 8 charge sectors with $D\text{max} = 27$, yielding a total bond dimension of $D_{\text{total}} \approx 700$.

The second class of methods instead starts from an approximation of the time evolution operator $\exp(-iH\delta)$ for a certain time step $\delta$. This approximation is provided in terms of a quantum circuit, or, more generally, an MPO, and can be obtained from e.g. a Suzuki-Trotter decomposition[6,20,21] or a cluster expansion[16,22]. The resulting MPO is then applied to the current state, encoded as MPS, followed by a bond truncation[1]. With a (low-order) Suzuki-Trotter decomposition, the MPO bond dimension can remain low, but feasible time steps $\delta$ are also very small. With the cluster expansion, it is easier to reach larger $\delta$, at the cost of a higher MPO bond dimension. It is therefore infeasible to apply this MPO to an MPS and truncate directly according to the Schmidt values due to prohibitive memory constraints or time complexity considerations. In this case thus, our method is indispensable.

---

[1] Note that methods based on Krylov subspaces or Taylor expansions of the evolution operator, which are common for time-evolving finite MPS, do not work in the thermodynamic limit because they are not extensive.

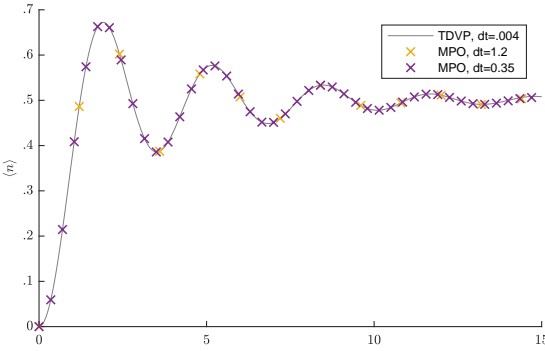

FIG. 2: Time evolution of the occupation number for the Néel state evolved with the XXZ Hamiltonian with $\Delta = 1/2$. We show results for different time steps for the MPO cluster expansion. The gray line is a reference result obtained with TDVP with very small time step. We have made explicit use of the $U(1)$ symmetry, and fixed the total bond dimension to $\chi = 994$.

We illustrate this usage by evolving the Néel state with the XXZ Hamiltonian.,

$$H_{\text{XXZ}} = \sum_i S_i^x S_{i+1}^x + S_i^y S_{i+1}^y + \Delta S_i^z S_{i+1}^z,$$

where $S_i^\alpha$ the spin-1/2 operators at site $i$ and we choose $\Delta = 1/2$. This problem is closely related to the one considered in Ref. 23 asserting the supremacy of quantum simulators. We have exploited the $U(1)$ symmetry of the system and used an MPS bond dimension of 994. The MPO bond dimension is 21, which enabled an accurate time step of up to $dt = 1.2$. In Fig. 2 we show the occupation number as a function of time, and benchmark it with a simulation with the TDVP algorithm with much smaller time steps.

*Power method for transfer matrices.*—Let us now consider the calculation of an MPS fixed point of an MPO transfer matrix by way of the power method: In each iteration we apply the MPO and truncate the bond dimension, until the MPS converges to a fixed point. Power methods have been used for computing transfer fixed points where the local singular-value truncation was adopted in each iteration[9], but here we use our variational truncation. In contrast to the former, the fixed point of our variational-truncation approach is, in fact, a variationally optimal MPS in the sense that it optimizes the leading eigenvalue for hermitian transfer matrices. Indeed, in the fixed point of this power method, the top-layer MPS in the fixed-point equations [Eqs. (17)-(19)] should be the same as the down-layer, and the equations reduce to the usual fixed-point equations of the VUMPS algorithm (which is variationally optimal for hermitian transfer matrices). Hence, both approaches share at least the same fixed point, which is not true if truncation based on singular values is performed.

For hermitian transfer matrices the performance of a

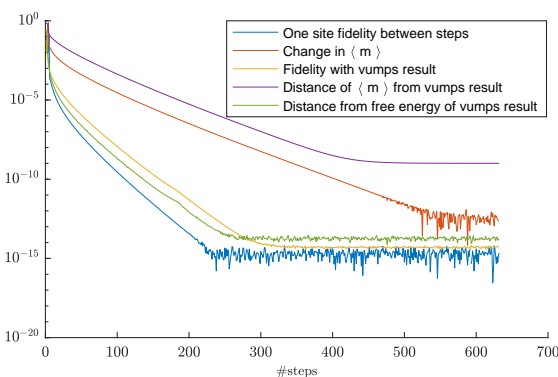

FIG. 3: Different error measures to determine the convergence of the vomps based power method approach to find the MPS fixed point of the MPO transfer matrix of the antiferromagnetic Ising model at inverse temperature $\beta = 1.01\beta_c$. From top to bottom in the legend, we show (1) the one site fidelity between site 1 and site 2 an iteration later, (2) the change in the local magnetization after an iteration, (3) fidelity with the sublattice rotated VUMPS result, (4) difference of the local magnetization with the one from the VUMPS result, (5) difference of the free energy with the one from the VUMPS result.

power method is inferior to that of the Krylov-inspired VUMPS algorithm, but it is very useful in cases of spatial symmetry breaking where the fixed point alternates between different MPSs or for non-Hermitean MPOs. We illustrate this case by studying the MPO transfer matrix of the classical antiferromagnetic Ising model on the square lattice (Fig. 3). In the (low-temperature) symmetry-broken phase, we find that the power method alternates between two MPSs that are the same up to a one-site translation. We look at some convergence criteria and also compare to the sublattice rotated ferromagnetic fixed point, found using VUMPS. Additionally we find that this technique allows for slightly better convergence of the fixed point MPS than VUMPS, as can be be seen from the stagnation of the fourth (purple) curve and the continued convergence of the second (red) curve and in Fig. 3.

*Dynamical growing of bond dimension.*—Our variational-truncation approach is particularly useful as a way of enlarging the bond dimension of an MPS when simulating time evolution or computing fixed points of transfer matrices. With respect to the former, the most persistent critique to the TDVP algorithm revolves around the fact that it projects the time evolution on the manifold of MPS with a fixed bond dimension, and that it is impossible to grow the bond dimension during the evolution. Our variational algorithm is not confined to a manifold of fixed bond dimension, because we can choose the bond dimension at each time step. We believe that a 'hybrid' between TDVP and our current scheme can provide a good way of simulating time evolution variationally using MPS where the amount of entanglement increases through time.

For fixed points of transfer matrices we can exploit our fidelity optimization in a similar way. We imagine the situation in which we have found a fixed-point MPS of a certain bond dimension, and we wish to find a better MPS of larger bond dimension. We can now use the previous MPS to construct an initial guess, apply the transfer matrix to this MPS, and then truncate to an MPS of the desired bond dimension using the equations above [Eqs. (17)-(19)]. The resulting MPS is already a more accurate approximation of the desired state than the previous one, and thus makes an excellent initial guess for running a new fixed-point algorithm at this higher bond dimension. This is especially useful in the context of PEPS algorithms, where the fixed point calculation of the PEPS double layer is the main bottleneck.

*Conclusions.*—We have discussed a method for approximating a uniform and infinite MPS by an MPS of lower bond dimension in a way that is variationally optimal. This method is proven most useful if the MPS being approximated has some substructure, like being made up of an MPO times and MPS. In this case the method has lower complexity and requires less memory than standard alternatives. We illustrate this with time evolution using an MPO that approximates the evolution operator, a power method for finding transfer matrix fixed points, and dynamical growing of bond dimension.

The generalization of this method to the (2+1)-dimensional case can easily be envisioned, and would be interesting to investigate. An algorithm that variationally determines a PEPS approximation of some other PEPS—perhaps a projected entangled-pair operator (PEPO) times a PEPS—can readily be devised based on the algorithm in Ref. 24. The uses of such a method would be identical to the ones presented here: performing accurate and reliable time evolution, a power method for determining fixed points of non-hermitian PEPOs or PEPOs exhibiting spatial symmetry breaking, and growing of a PEPS bond dimension.

*Acknowledgements.*—This project has received funding from the European Research Council (ERC) under the European Unions Horizon 2020 research and innovation programme (grant agreement No 647905 – QUTE and No 715861 – ERQUAF) and from the Research Foundation Flanders (grant No G087918N).

* Electronic address: bavhecke.Vanhecke@UGent.be

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
