# Peer review of "Tangent-space methods for truncating uniform MPS"

_SciPost Physics Core_

## Round 1 · Referee Report · Anonymous (Referee 1) · 2020-11-3

Strengths

  1. The manuscript introduces a concrete algorithm to solve a relevant problem in the field.
  2. A few proof-of-principle numerical examples are addressed with the proposed algorithm.

Weaknesses

  1. A couple of the manuscript's claims are not given full justification, and some other omissions should be addressed (see below).
  2. In its current form, the manuscript is probably more useful for experts in the field. Including a few explanatory points can make it more accessible to a broader audience.

Report

In this paper, the authors present an algorithm which operates on the setting of uniform matrix product states (uMPS), a tensor network ansatz for translation invariant one-dimensional quantum states in the thermodynamic limit. The algorithm allows to variationally approximate a given uMPS with another one of lower bond dimension, thus effectively reducing the number of parameters. The authors motivate their research by the fact that such a truncation is an important part of a series of tensor network algorithms, and justify the importance of their results by a) comparison of the computational-theoretic complexity and the optimality of the results of their algorithm with a previous approach based on Schmidt value truncation, and b) application of their algorithm numerically to several examples.

Regarding the expectations for a SciPost Physics Core submission, the problem addressed is certainly of interest for the tensor network community, since the approximation of MPS by other MPS with lower bond dimension is instrumental in many algorithms, as the authors correctly point out. In my opinion, the methods, which are framed within the well-developed formalism of uMPS, are appropriate and the results are satisfactory.

It should be pointed out that the central part of the paper, the truncation algorithm, is already presented in some detail in Ref. 13 of the manuscript, "Tangent-space methods for uniform matrix product states", by a subset of the authors, which was published by SciPost Lecture Notes in 2019. On the other hand, the complexity analysis and the numerical examples are to the extent of my knowledge new. It is my understanding that this is a case of a comprehensive review coming out before one of the original research papers it draws from, and thus I expect this should suppose no obstacle for publication of the present submission. It is however up to the editor to evaluate the importance of this particular circumstance.

In my opinion, the manuscript satisfies all acceptance criteria to a reasonable degree. There are however a few spots where I believe definitions, logical steps or justifications are missing, and some more places where optional clarifications would be very beneficial for the reader. Hence I invite the authors to go through the attached list of requested changes (ordered by appearance in the manuscript). Succesfully addressing points 5, 7, 8, 11, 12, 13 and 14 would be enough, from my point of view, to guarantee a recommendation for publication, while the rest are aimed at improving the usefulness of the paper for a potential reader.

Requested changes

  1. The introduction of the cost function in terms of overlaps of uMPS could gain in clarity by briefly addressing the particular nature of these overlaps: Are they finite? Is there a hidden limit of finite MPS? Is the expression defined only formally? Why do we include the denominator, since it should be equal to one because of Eq. (2)?

  2. The form given for $\mathcal{P}_A$ corresponds to a projector on the subspace of tangent vectors that are orthogonal to $|\Psi(A)\rangle$. The latter causes the rhs of Eq. (7) to vanish identically. This, I would say, is sufficiently remarkable that the failure to mention it can leave the reader trying to reproduce the results in confusion. A brief explanation along the lines of page 21 in Ref. 13 would make things much clearer for the reader.

  3. It is easy to see why Eq. (9) is a sufficient condition for the gradient to vanish, but it is rather unclear from the paper why it should be necessary, i.e., why it follows from Eq. (7).

  4. The symbol $N$ for, presumably, the length of some undefined finite MPS appears in Eq. 13 without explanation. Nor is it explained if the limit should always exist (e.g. the matrices always have a single leading eigenvalue) or if it does so only for almost every tensor, in the sense of measure theory. The statement is thus intuitively understandable only for readers with some experience in the area.

  5. The authors present the iterative method without mentioning the initialization step. As a reader, I would assume the initialization is random or depends on the particular problem we are solving, and if that is the case it would be clearer to say it.

  6. I would say Eq. (16) is the norm of Eq. (9), not Eq. (7).

  7. Could it be that the cost of operation (20) is missing a square, as in $\mathcal{O}(\chi^3D^2d+\chi^2D^3d^2)$ instead of what is written? I have not been able to find a better contraction scheme.

  8. Regarding Fig. 1, there is no definition for the "variational error" in the Schmidt truncation case. Is it the size of the discarded singular values? Additionally, it would be useful to know how the actual fidelity changes iteration after iteration, instead of just plotting the variational error. Since the latter drops by several orders of magnitude to obtain a relatively small improvement in the fidelity per site, the plot as is gives the reader little information as to when it would be reasonable to stop iterating, i.e. what is a good value of $\eta$. Also, is the improvement in accuracy as tiny as it seems from the numerical value? For someone who is not used to thinking of fidelities per site, a bit of context as to whether the improvement is significant or not would be useful. Finally, for the sake of reproducibility, including the explicit Hamiltonian and the way the initial MPS was obtained, maybe in an appendix, would be helpful.

  9. In the comparison between time evolution with the proposed method and with TDVP (Fig. 2), it would be nice to give the reader a notion of the computational cost of both simulations (i.e. an estimate for the time it takes to run each of them in a given computer), to allow her to appreciate the magnitude of the improvement, for equally accurate results. Also, writing the Néel state more explicitly would help unfamiliar readers.

  10. It would be nice to indicate the reference for the superiority of Krylov-inspired VUMPS over power methods in the case of Hermitian transfer matrices.

  11. In Fig. 3, it is stated that fidelities are plotted, when, I understand, it is their differences from one that are shown.

  12. The authors should explain more explicitly, or give a reference for, the appearance of the sublattice rotated ferromagnetic fixed point, and why the comparison with the other fixed point they are computing is relevant.

  13. The comparison between the red and purple curves in Fig. 3 shows that the proposed truncation algorithm and VUMPS have different fixed points (very close fidelity-wise, but appreciably different in $\langle m\rangle$). The authors argue that this implies "better convergence" of the former, which I find unclear. If they mean that the fixed point they obtain is more accurate, they should motivate why that is the case, since a priori a similar plot could be obtained in the case were the VUMPS fixed point is better. If they mean something else, a more explicit rephrasing would be necessary.

  14. In the conclusions, the author state that "This method is proven most useful if the MPS being approximated has some substructure, like being made up of an MPO times and MPS. In this case the method has lower complexity and requires less memory than standard alternatives." However, the manuscript has analyzed only that particular case, and not any other to compare it with. Thus, for it to make sense to say something like that, I would expect the authors to have performed a similar analysis of computational cost for the case when the MPS being approximated has no substructure, and found that the improvement from using their algorithm is not as significant, in which case, they should probably report it. That said, I have quickly checked it and it seems to me that the computational cost advantage is still there. If that were to be the case, then there is no apparent reason to say the new method is "most useful" in the particular case, unless what is meant is that the latter comes up more often in practice.

  • validity: high
  • significance: good
  • originality: good
  • clarity: ok
  • formatting: perfect
  • grammar: perfect

Author:  Bram Vanhecke  on 2021-01-13  [id 1147]

(in reply to Report 1 on 2020-11-03)

We would like to thank the referee for the studious analysis of our work. We have updated the manuscript according to his/her comments. Here, we address the most relevant points. It is correct that this method was already touched upon in a lecture notes on tangent-space methods, but there it only served as an algorithmic steppingstone to the vumps algorithm and was mentioned without benchmarks or possible applications in tensor networks. In this paper, we want to give a self-contained exposition, and show the power and usefulness. In response to the comments 1. We have clarified this. 2. We have rewritten the section pertaining to the tangent space projector in accordance with your notes. 3. The two expressions are, in fact, equivalent. We have included a sentence below Eq. (17) to elucidate this point. 4. This was corrected in the text. 5. We have added that we start from a random MPS (in practice, there could be better choices, of course). 6. Indeed. 7. Indeed, we have corrected this mistake. 8. The definition of the ‘variational error’ was pointed to as clarification. The VUMPS algorithm was used for getting this ground state, we now state this and give a reference discussing this model in more detail. 9. Each individual step in our time-evolution scheme is more costly than taking a TDVP step, though fewer steps need to be taken. We believe it best suited to increase bond dimension in a natural way in a TDVP scheme. We believe that the Neel state is a well-known concept in the literature. 10. Reference included 11. We have corrected this. 12. The antiferromagnetic Ising model is related to the ferromagnetic Ising model by a simple sublattice rotation, so the fixed points can be easily related as well. We have included an explicit note specifying what this sublattice rotation amounts to, but we believe it would lead us too far to explicitly work out in detail why this procedure is correct. 13. Indeed this was a strange result, that we overzealously interpreted as showing an improved convergence as compared with the VUMPS result. We have tried to reproduce the results to do some more analyses, to properly address the referee’s question. However, we never saw this non-convergence of the magnetisation first reported again. We therefore removed the (correctly identified) speculative part in the text, and updated Fig.3. 14. We note that our first figure is providing such a test case, where we show that our variational approach performs slightly better, but in practice the local truncation is a very good and simple heuristic. It is only when we have a substructure, and the computational cost is actually lower, that our variational approach is deemed to be particularly useful.

---

## Round 1 · Referee Report · Johannes Hauschild (Referee 2) · 2020-11-5

Strengths

1) The paper provides a full description of an algorithm to approximate an uMPS by another uMPS with lower bond dimension, with all necessary definitions to allow a straight-forward implementation. 2) It demonstrates the usefulness with examples of possible applications.

Weaknesses

1) The axis labels of the figures are a bit unclear.

Report

In this work, the authors discuss an algorithm for the compression of an uniform MPS, a tensor network ansatz representing a strictly translation invariant state in an infinite 1D lattice. Tangent space methods used allow to keep the ansatz structure and hence provide a very elegant and mathematically appealing way to directly work in the thermodynamic limit. The compression of a given MPS with high bond dimension by one with a lower one is one of the most fundamental building blocks for tensor network calculations.

In my opinion, the paper meets all the criteria for Scipost Physics core.
I therefore recommend the publication once the minor changes requested below (and in the previous report) have been addressed.

Requested changes

1) There's a missing bracket in the line following Eq. 4. 2) Eq. 15 should have a $V$ instead of an $U$. 3) The line after Eq. 8 should contain $\vert\Psi(M)\rangle$. 4) I don't get why there appears a $\lambda$ in Eq. 16. Is it supposed to be there? 5) What is the definition of the "variational error" plotted in Fig. 1? 6) What is the "occupation number" plotted in Fig. 2? Naively, I would have identified the total Sz with 'n' after a Jordan-Wigner mapping, but this is a conserved quantity. 7) The caption of Fig. 3 calls the talks about a "vomps" based power method. This is mentioned nowhere else, so it's unclear that it refers to the method described in the main text. 8) The authors suggest to use time evolution based on application of $U=\exp(iHdt)$ as an MPO with the presented compression method to dynamically grow the bond dimension, possibly as a hybrid scheme with TDVP. However, the presented algorithm takes the desired bond dimension of the compressed uMPS as an input variable, and one needs to guess the necessary bond dimension a priori. Is there a scheme to automatically determine the necessary bond dimension for a given error tolerance?

  • validity: high
  • significance: good
  • originality: high
  • clarity: good
  • formatting: excellent
  • grammar: perfect

Author:  Bram Vanhecke  on 2021-01-13  [id 1146]

(in reply to Report 2 by Johannes Hauschild on 2020-11-05)

We thank the referee for the positive feedback. We have changed the paper to accommodate all the proposed changes and corrections and have improved the exposition so that the answers to the questions posed by the referee can now hopefully be obtained or inferred from the text.
In response to the last point of the referee, we note that there is a clear heuristic for this: If the smallest Schmidt-value for the MPS is, at a certain point in the time-evolution, above a certain threshold or tolerance, one can choose to increase the bond dimension.

---

## Round 1 · Referee Report · Anonymous (Referee 3) · 2020-11-16

Strengths

  1. Derivation of variational truncation method for infinite, uniform MPS
  2. Applicable also to approximating outcome of MPO-application to MPS
  3. Detailed algorithmic formulation of the presented method
  4. Discussion of various test cases

Weaknesses

  1. Very rough and to some extend misleading presentation of the seemingly central equality in the derivation
  2. Unclear numerical setups in the presented applications
  3. It looks like in the conclusion, there are is an unclear statement

Report

In this manuscript the authors present and discuss a method to variationally compress an infinite and uniform MPS. The method itself was already introduced by some of the authors in the context of doi: 10.21468/SciPostPhysLectNotes.7, but it has not been analyzed in more detail, yet. For that reason I understand this manuscript as a detailed benchmark paper demonstrating the implementation and capabilities of the presented algorithm. Having said that I will comment only briefly on the method's presentation which, in my opinion, is very short and rather misleading at the central point (namely the transition from eqn. 7 and 9). If the manuscript is supposed to address a non-expert-audience (w.r.t. iuMPS) I'd suggest to extend the discussion so that it becomes clear, that $\mathcal P_A \vert \Psi(A) \rangle \equiv 0$ implying the subsequent conclusions. However, I guess it'd also be sufficient to remove this explanatory part and refer to 10.21468/SciPostPhysLectNotes.7 where the technical details are explained more transparently.

The presentation of the algorithm itself is done comprehensively and very detailed. In this part of the manuscript, the generalization to MPO-MPS-products is introduced, which is a new methodical aspect, as far as I know. There seems to be a typo in the given asymptotic scaling of the computational costs (missing square at the physical-leg dimension). However, if this is not the case, it'd be more than noteworthy what the employed contraction scheme looks like.

The provided examples cover various problem settings. At first the authors consider the truncation of a SU(2)-symmetric ground-state of a $S=4$-Heisenberg model. The truncation ratio is impressive, given the achieved accuracy measured in terms of the variational error $\epsilon$. However, in the caption of Fig. 1, the authors mention a variational error obtained from a local singular-value truncation. As far as I know, there is no variational error in singular-value-decomposition (SVD) based truncation schemes and an explanation or definition of the mentioned quantity would be helpful. Also, in the main text, the authors describe only a slightly better state after convergence of the variational approach, compared to the SVD based truncation scheme which seems to be a contradiction to the rather large error, shown in Fig. 1. Again, a clarification of the quantity to which the data is compared to would be very helpful. Also, a more detailed specification of the numerical parameters would help to value the success of the truncation (model's couplings, how is the ground state obtained in first place?). As a second example, the authors discuss the time evolution of a Neel state w.r.t. a XXZ-Hamiltonian. They compare the time evolution of a local observable, probably the expectation values $\langle S^z_j \rangle$, obtained either by time-evolving using a TDVP stepper (with very small time step) or an MPO stepper obtained from a cluster expansion according to Ref [16], with a rather large time step. Again, the numerical details are incomplete rendering the interpretation of the presented results very difficult. For instance, the initial state is supposed to be a Neel state so it is highly important to discuss the numerical specifications of the exploited TDVP solver (has there already been used the dynamical growing of the bond dimension, discussed later?). Furthermore, is the bond dimension $\chi=994$ hold fix throughout the total time-evolution or is it permitted to grow upto $\chi=994$ (if so, how is the growing realized, if not, how is the initial state prepared)? As a third example the authors present an application to obtain transfer matrices by means of the power method. The demonstrated convergence compared to VUMPS data is convincing w.r.t. to the better description of the sublattice-symmetry broken MPS obtained by the power method. exploiting the presented variational truncation method. Finally, the authors discuss a scheme to dynamically grow the bond dimension of an MPS. To me, this seems to be a very strong point in their manuscript. In particular, resolving the bond-dimension constraints of TDVP would be remarkable and the question arises, if the authors may already have some data suggesting their claims. A similar situation holds for the discussed growing of the bond dimension when calculating fixed points of transfer matrices.

In the conclusion, the authors mention that the variational method is most useful, if the MPS has some substructure (i..e. being obtained from applying an MPO to an MPS). However, Fig. 1 seems to imply, that the variational truncation method is also superior to the SVD-based truncation, at least if I only compare the pure outcome of the fidelity comparison.

Some further questions/typos I stumbled over are:

  1. p.2, below eqn. 7: $\vert \psi_A \rangle$ has not been defined before
  2. Ref. 22 is not based on a Cluster expansion
  3. p.4, Fig. 2: What is $n$?
  4. p.4, Caption of Fig.3: What is vomps?

In summary, the manuscript presented by the authors presents various applications of their developed variational truncation method for infinite and uniform MPS. The chosen examples are representative for a broad class of problems and provide a good overview on the potential of the method. The very detailed algorithmic presentation helps to implement this method very quickly. However, in my opinion the discussion of the different applications is too rough and there there are various technical details missing that are required to appropriately value the numerical properties of the presented method. Also, in the summary there appears to be a statement that is a bit counterintuitive when compared to the presented data in Fig. 1. I'd suggest to resolve the mentioned issue in the introduction of the method and to extend the technical details in the comparison section.

Requested changes

  1. Either extend explanation of eqn. 6-9 or remove the discussion and refer to Ref. [13], just quoting the outcome here
  2. Resolve open question about the computational cost of MPO-MPS-application (is it only a typo or did the authors use a particular contraction scheme?)
  3. Provide more details for all presented benchmark calculations so that they are reproducible
  4. Please clarify: How do the authors come to the conclusion about the method being most useful when compressing MPO-MPS-products?

  • validity: good
  • significance: high
  • originality: good
  • clarity: ok
  • formatting: good
  • grammar: excellent

Author:  Bram Vanhecke  on 2021-01-13  [id 1145]

(in reply to Report 3 on 2020-11-16)

We would like to thank the referee for her/his careful reading and for the detailed and informative report. We have changed the manuscript according to her/his comments and hope that most questions are answered through our revisions. Here, we give a few responses to the most pertinent points.
It is correct that this method was already touched upon in a lecture notes on tangent-space methods, but there it only served as an algorithmic steppingstone to the VUMPS algorithm and was mentioned without benchmarks or possible applications in tensor networks. In this paper, we aim to provide a self-contained exposition of this algorithm, and illustrate its power and usefulness.
We agree that the transition between eq. (7) and (9) was misleading and have therefore rewritten this part to be more transparent and accurate.
The computational cost was, indeed, not correct; we have corrected this.
The variational error we define is not dependent on any specific algorithm, and can be defined for any MPS. Therefore, we can also compute it for the MPS that results from performing the standard local-truncation approach. It is quite surprising how well local truncation performs in terms of fidelity, given that the associated variational error is still quite large; this, of course, justifies the wide use of local truncation in MPS algorithms. We comment in the paper on this surprisingly good result that local truncation provides.
Indeed, the TDVP simulation was left a bit mysterious in the previous version. We have now included the references for the heuristics of extending the bond dimension. We have chosen not to explain this in the paper itself, because TDVP only serves as a consistency check here.
We have included a sentence in the conclusion stating that our method performs slightly better in terms of accuracy than the standard approach in the community, and that it is probably most useful in the case of the MPS having a substructure because of the reduced computational cost in that case.

---

## Editorial Decision

resubmitted